ecology

phytochemical, emergent infectious disease, caffeine, bumblebee, microsporidia, sainfoin

**Author for correspondence:**
Arran J. Folly
e-mail: arran.folly@apha.gov.uk

†Present address: Virology Department, Animal and Plant Health Agency, Surrey, UK.

# Agri-environment scheme nectar chemistry can suppress the social epidemiology of parasites in an important pollinator

Arran J. Folly[1,†], Hauke Koch[2], Iain W. Farrell[2], Philip C. Stevenson[2,3] and Mark J. F. Brown[1]

[1]Centre for Ecology, Evolution and Behaviour, Department of Biological Sciences, School of Life Sciences and the Environment, Royal Holloway University of London, Egham, UK
[2]Royal Botanic Gardens, Kew, UK
[3]Natural Resources Institute, University of Greenwich, Kent, UK

AJF, 0000-0003-0106-2185; PCS, 0000-0002-0736-3619; MJFB, 0000-0002-8887-3628

Emergent infectious diseases are one of the main drivers of species loss. Emergent infection with the microsporidian *Nosema bombi* has been implicated in the population and range declines of a suite of North American bumblebees, a group of important pollinators. Previous work has shown that phytochemicals found in pollen and nectar can negatively impact parasites in individuals, but how this relates to social epidemiology and by extension whether plants can be effectively used as pollinator disease management strategies remains unexplored. Here, we undertook a comprehensive screen of UK agri-environment scheme (AES) plants, a programme designed to benefit pollinators and wider biodiversity in agricultural settings, for phytochemicals in pollen and nectar using liquid chromatography and mass spectrometry. Caffeine, which occurs across a range of plant families, was identified in the nectar of sainfoin (*Onobrychis viciifolia*), a component of UK AES and a major global crop. We showed that caffeine significantly reduces *N. bombi* infection intensity, both prophylactically and therapeutically, in individual bumblebees (*Bombus terrestris*), and, for the first time, that such effects impact social epidemiology, with colonies reared from wild-caught queens having both lower prevalence and intensity of infection. Furthermore, infection prevalence was lower in foraging bumblebees from caffeine-treated colonies, suggesting a likely reduction in population-level transmission. Combined, these results show that *N. bombi* is less likely to be transmitted intracolonially when bumblebees consume naturally available caffeine, and that this may in turn reduce environmental prevalence. Consequently, our results demonstrate that floral phytochemicals at ecologically relevant concentrations can impact pollinator disease epidemiology and that planting strategies that increase floral abundance to support biodiversity could be co-opted as disease management tools.

## 1. Introduction

Two prominent drivers of biodiversity loss are emerging infectious diseases (EID) [1–7] and the reduction of natural habitat [8–13], often as a direct consequence of intensive agriculture [14–16]. One important approach to mitigate the negative impact of agriculture on biodiversity has been the development of agri-environment schemes (AES) [17,18] and conservation reserve programmes (CRP) [19]. In both AES and CRP, prescriptions are set out that increase floral abundance and diversity to conserve and enhance broader biodiversity, and the ecosystem services, such as pollination, that it supplies [17–19]. Such approaches have been shown to increase the abundance, diversity and persistence of beneficial organisms, such as

pollinators, in agricultural environments [20–23]. However, pollinators are also threatened by EIDs [24,25], and AES prescriptions may indirectly act as hubs that amplify disease in these populations [25–27]. Floral rewards, the main attractant for pollinators, contain secondary metabolites [28–30]. These phytochemicals can have antimicrobial properties [31] and may therefore have positive effects on pollinator disease by controlling parasites and pathogens, mitigating or overwhelming the opportunities for parasite transmission that they also provide [32–36]. Consequently, strategies that increase floral abundance and diversity, if designed to include floral mixes that incorporate high nutritional and medicinal value, may improve pollinator health and therefore could be co-opted to manage wildlife diseases.

Bumblebees are key global crop pollinators, providing up to 80% of pollination services for some crops [37–39]. However, these charismatic insects are threatened by EIDs [40,41]. More specifically, disease spillover has been identified between the European honeybee (Apis mellifera) and UK bumblebees [25,42], and the global movement of commercial bumblebees for crop pollination has introduced novel pathogens to naive bumblebee communities in both North and South America [24,43]. One such emergent pathogen is the microsporidian Nosema bombi [44], which has been implicated in the population and range declines recorded in a suite of North American bumblebees over the past four decades [24,45]. Bumblebees are unable to recognize and remove N. bombi infected brood [46], suggesting that social immunity [47] is insufficient to adequately prevent disease transmission within a colony. Given that animals consume naturally occurring, bioactive compounds in their diets [48–50], inspiration for novel medications to combat endemic and emergent parasitic infections may be drawn from an animal's pre-existing diet [51–53], as has been shown to be the case for individual bumblebees [34–36]. However, such studies have yet to address the social epidemiology of pollinator parasites, which is key to their success in highly social insects [54].

Here, we screen flowers used in UK-based AES, targeted at bumblebees, for phytochemicals in both the pollen and nectar using a combination of high-performance liquid chromatography and mass spectrometry (LC-MS). Caffeine was identified in the nectar of sainfoin (Onobrychis viciifolia), a constituent of UK AES and a major global crop [55]. We then designed a range of experiments using commercial bumblebee colonies to determine if consumption of caffeine can impact N. bombi infection in individual bumblebee workers, both prophylactically and therapeutically. We then used colonies reared from wild bumblebee queens to investigate whether caffeine can impact social, and by extension environmental epidemiology. Given that phytochemicals impact disease in individual bumblebees, we predict that if caffeine has a negative impact on N. bombi in individuals it will also impact social epidemiology. Our results show that caffeine did reduce infection intensity and prevalence in individual bumblebees and that this impact was reflected in our colony-level experiments.

## 2. Results

### (a) Identification and analysis of phytochemicals from agri-environment scheme plants

Nine species of flowers were identified from AES seed mixes prescribed by Natural England (2017) as potential bumblebee forage (electronic supplementary material, table S1). Forty-one and 27 phytochemicals (inclusive of isomers) were identified, respectively, from pollen and nectar sampled from the nine AES species, using LC-MS and high-resolution electrospray ionization mass spectrometry (HR-ESI-MS) (electronic supplementary material, table S2 and table S3, respectively). Consequently pollinators such as bumblebees, are repeatedly exposed to a range of phytochemicals, which may be bioactive, during foraging bouts in AES-enhanced landscapes. We identified the alkaloid caffeine in the nectar of sainfoin (Onobrychis viciifolia) (concentration range 0.35–200 µM, from one pooled sample in 2017 and three pooled samples from 2019) using a retention time (Rt) of 6.11 min and a mass-to-charge ratio (m/z) of 195.09, which has known biological activity [56,57]. For all subsequent experimental procedures, we selected 200 µM of caffeine to investigate its impact on N. bombi. While this was the highest concentration recorded in our analyses, from a limited number of sampling points and locations, it was representative of natural caffeine concentrations reported elsewhere in floral nectar [58,59].

### (b) Caffeine negatively impacts Nosema bombi infection in Bombus terrestris workers

In the prophylactic bioassay, 82 workers successfully eclosed, of which 36 had N. bombi infections (control = 22, caffeine = 14), resulting in an overall infection success of 44%. There were no significant differences between the infection prevalence in our control or experimental groups ($\chi^2 = 1.433$, $p = 0.231$), indicating that caffeine had not affected parasite prevalence. By contrast, caffeine did have a significant prophylactic effect by reducing N. bombi infection intensity in eclosed workers (linear mixed-effects (LMM), $F_{1,28} = 15.33$, $p < 0.001$; figure 1a). The covariates thorax width (LMM, $F_{1,28} = 3.75$, $p = 0.06$), faeces volume (LMM, $F_{1,28} = 0.65$, $p = 0.4$) and the random effect colony ($p = 0.27$) all had no significant effect on N. bombi infection intensity.

In the therapeutic bioassay, 101 workers successfully eclosed, of which 39 had N. bombi infections (control = 25, caffeine = 14), resulting in an overall infection success of 39%. Again, there was no significant difference between the infection prevalence in our control or experimental groups (caffeine $\chi^2 = 2.384$, $p = 0.123$). However, as with our prophylactic bioassay, caffeine had a significant therapeutic effect by reducing N. bombi infection intensity in eclosed workers (LMM, $F_{1,31} = 4.97$, $p = 0.032$; figure 1b). The covariates thorax width (LMM, $F_{1,31} = 2.49$, $p = 0.123$), faeces volume (LMM, $F_{1,31} = 1.87$, $p = 0.181$) and the random effect colony ($p = 0.193$) all had no significant effect on N. bombi infection intensity.

### (c) Caffeine significantly impacts the epidemiology of Nosema bombi in Bombus terrestris colonies

Caffeine treatment significantly reduced the prevalence of N. bombi in colonies (LMM, $F_{1,851} = 18.04$, $p < 0.001$; figure 2). Prevalence varied across colonies ($p = 0.001$), but the covariates brood cohort (LMM, $F_{1,851} = 3.36$, $p = 0.06$), number of inoculated larvae (LMM, $F_{1,851} = 0.02$, $p = 0.90$), number of days since inoculation (LMM, $F_{1,851} = 0.23$, $p = 0.63$) and the random effect forager ($p = 0.06$) all had no significant effect on N. bombi infection prevalence. Caffeine had a similar negative impact on N. bombi infection intensity (LMM, $F_{1,851} = 34.8$,

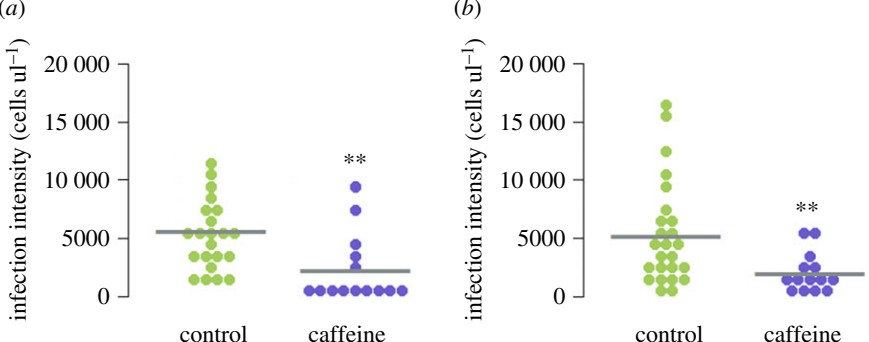

**Figure 1.** Beeswarm plots of *N. bombi* infection intensities in adult *B. terrestris* workers that had been fed caffeine prophylactically (*a*) and therapeutically (*b*). The sample mean has been marked with a grey bar and significant differences have been marked with a double asterisk. Caffeine was found to have both a significant prophylactic and therapeutic effect on *N. bombi* infection intensity in *B. terrestris* workers. (Online version in colour.)

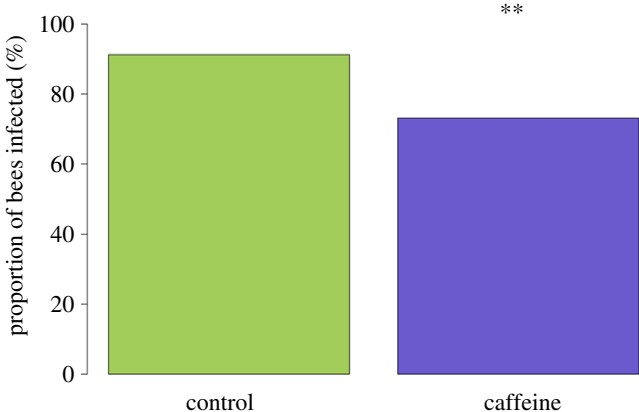

**Figure 2.** *Nosema bombi* infection prevalence in adult bumblebees (*B. terrestris*) for both control ($n = 438$) and caffeine ($n = 272$) treatments. Treatment with caffeine significantly reduced *N. bombi* infection prevalence (marked with a double asterisk). (Online version in colour.)

$p < 0.001$; figure 3). In addition, there was a significant negative interaction between cohort and treatment on *N. bombi* infection intensity ($F_{1,851} = 14.2$, $p < 0.001$), meaning that the younger the bumblebee, the greater the impact caffeine treatment had on reducing *N. bombi* infection intensity (figure 4). By contrast, the covariates number of inoculated larvae (LMM, $F_{1,851} = 0.24$, $p = 0.6$), number of days since inoculation (LMM, $F_{1,851} = 0.001$, $p = 0.9$) and the random effects forager ($p = 0.06$) and colony ($p = 0.51$) all had no significant effect on *N. bombi* infection intensity. Finally, the covariates treatment (LMM, $F_{1,205} = 0.937$, $p = 0.02$) and cohort (LMM, $F_{1,205} = 2.294$, $p < 0.001$) were found to have a significant impact on forager infection prevalence. In essence foragers, especially younger foragers, from our caffeine-treated colonies were less likely to have *N. bombi* infections. By contrast, the infection intensity of infected foragers was not impacted by treatment (LMM, $F_{1,205} = 1.509$, $p = 0.14$), but interestingly cohort did have a significant negative impact on forager infection intensity (LMM, $F_{1,205} = 3.061$, $p < 0.001$), showing that infected younger foragers had lower infection intensities (figure 5). In both forager models faeces volume (prevalence LMM, $F_{1,205} = -1.505$, $p = 0.13$, infection intensity LMM, $F_{1,205} = 0.342$, $p = 0.7$), number of larvae inoculated (prevalence LMM, $F_{1,205} = -0.242$, $p = 0.8$, infection intensity LMM, $F_{1,205} = 01.323$, $p = 0.19$) and colony (prevalence LMM $p > 0.05$, infection intensity LMM $p > 0.05$) had no impact on infection prevalence or intensity.

## (d) No impact of caffeine on bumblebee colony growth and reproduction

The covariates treatment (LMM, $F_{1,18} = 2.45$, $p = 0.13$), number of brood inoculated (LMM, $F_{1,18} = 0.77$, $p = 0.3$) and number of days since inoculation (LMM, $F_{1,18} = 0.004$, $p = 0.4$) had no effect on the number of workers produced in *B. terrestris* colonies. Similarly, the covariates treatment (LMM, $F_{1,18} = 0.96$, $p = 0.3$), population size (LMM, $F_{1,18} = 2.13$, $p = 0.17$), number of brood inoculated (LMM, $F_{1,18} = 0.04$, $p = 0.85$), days since inoculation (LMM, $F_{1,18} = 0.032$, $p = 0.25$) and the random effect colony ($p = 0.82$) had no effect on the production of sexual castes.

## 3. Discussion

Here, we show that caffeine, which may be encountered by bumblebees in AES landscapes, can negatively impact both the individual and social epidemiology of a key bumblebee pathogen, *N. bombi*. Consumption of caffeine reduced the overall infection prevalence and intensity of *N. bombi* infections in *B. terrestris* colonies reared from wild-caught queens, while simultaneously having no negative impact on colony development. Consequently, our results indicate that schemes that increase floral abundance and diversity, such as AES and CRP, may have unexplored benefits as they have the potential to be co-opted as disease management tools to further improve pollinator health in the field.

AESs and CRP aim to increase floral abundance and diversity in agricultural landscapes to benefit wider biodiversity. Our results highlight that pollinators, such as bumblebees, are repeatedly exposed to a diverse suite of phytochemicals, which may have bioactive properties, in pollen and nectar, during foraging bouts [30]. Understanding how bioactive phytochemicals impact pollinator health should be an essential component for measuring the effectiveness of schemes, such as AES and CRP, which aim to support biodiversity. Here, we identified caffeine in the nectar of sainfoin (*Onobrychis viciifolia*), a component of UK AES, and showed that it negatively impacts the infection success and epidemiology of the microsporidian *N. bombi* in wild caught and reared *B. terrestris* colonies. Consequently, caffeine forage has the potential to mitigate the impact of *N. bombi* disease in wild bumblebees, both where the pathogen is endemic and in areas where it is emerging into naïve populations.

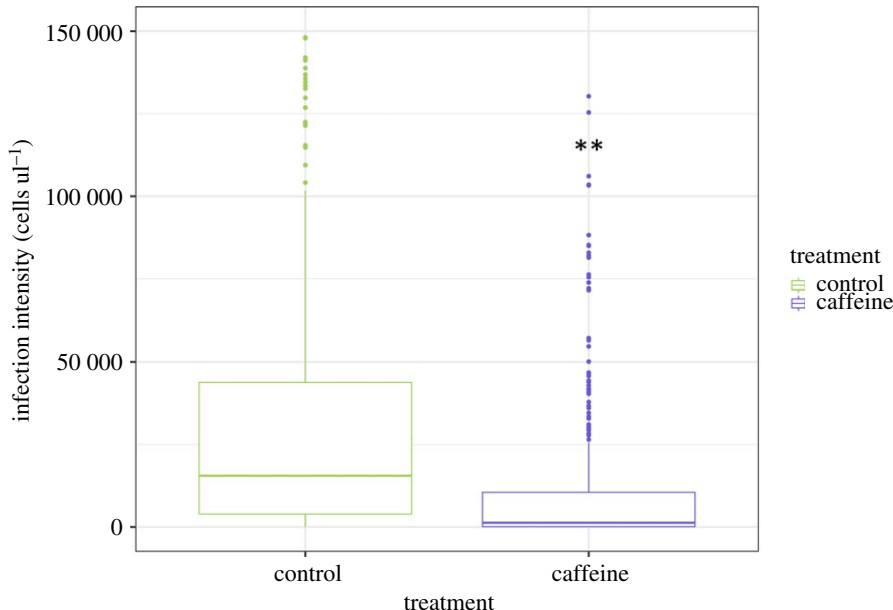

**Figure 3.** Treatment with caffeine significantly reduced *N. bombi* infection intensity. Box plot of *Nosema bombi* infection intensity in adult bumblebees (*B. terrestris*) for both control (*n* = 438) and caffeine (*n* = 272) treatments. Significant differences between treatments are shown with a double asterisk. (Online version in colour.)

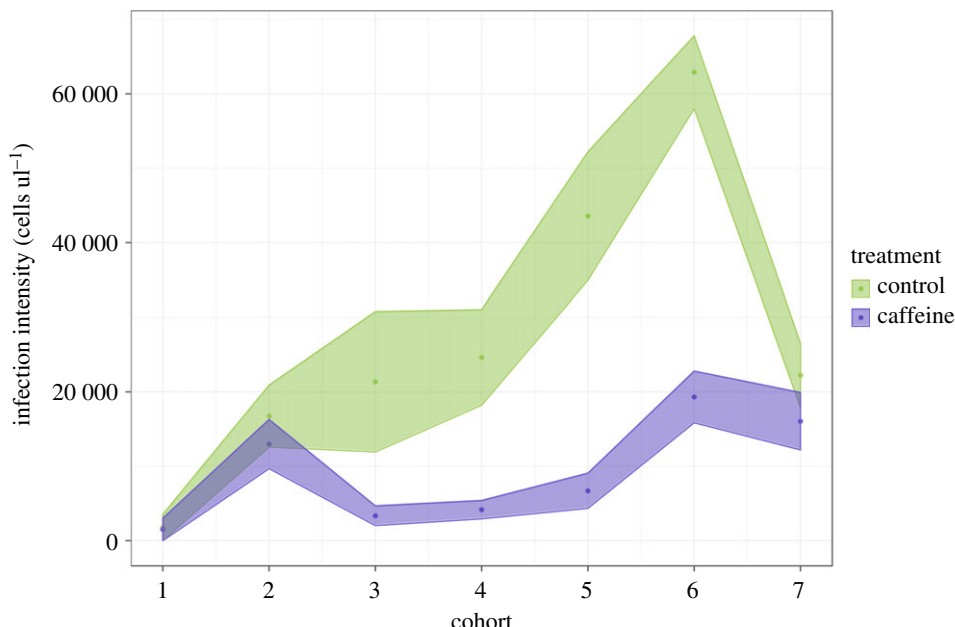

**Figure 4.** *Nosema bombi* infection intensity across *B. terrestris* brood cohorts. Each brood cohort is a representation of colonies reproductive output over two weeks, with the first cohort representing the first batch of brood produced. Hence, at sampling, later cohorts contained younger bumblebees. Cohort and treatment significantly impacted *N. bombi* infection intensity in adult bumblebees. Caffeine feeding reduced *N. bombi* infection intensity and earlier cohorts across both treatments had lower infection intensities. Shaded areas represent mean ± s.e.m. (Online version in colour.)

While our assessments of caffeine bioactivity were at the highest concentration likely to be encountered in wild *O. viciifolia*, this is still within naturally occurring nectar concentrations reported from other caffeine-producing plants [58]. In addition, caffeine is a common nectar constituent [59] which can suppress latent virus infection and reduce *N. ceranae* spore intensity in honeybees [56,57]. Consequently, our experimental paradigm represents an ecologically relevant interaction between pollinators, their pathogens and forage plants. However, we would note that further work is required to assess the variation across different cultivars and wild species of caffeine-producing plants [60].

Emergent infection with *N. bombi* has been implicated as a principal driver of contemporary bumblebee declines across North America [24,45]. Our results show that caffeine reduced the overall infection intensity of *N. bombi in vivo*, in individual bumblebees that were treated as larvae, in both prophylactic and therapeutic treatments. Both of these treatment methods are recognized in preventative medicine and are biologically realistic within the bumblebee–*Nosema* system. As infection by *N. bombi* may occur at any time across the lifecycle of bumblebee colonies, caffeine and potentially other bioactive secondary chemicals [36], may interact either prophylactically or therapeutically to prevent or inhibit

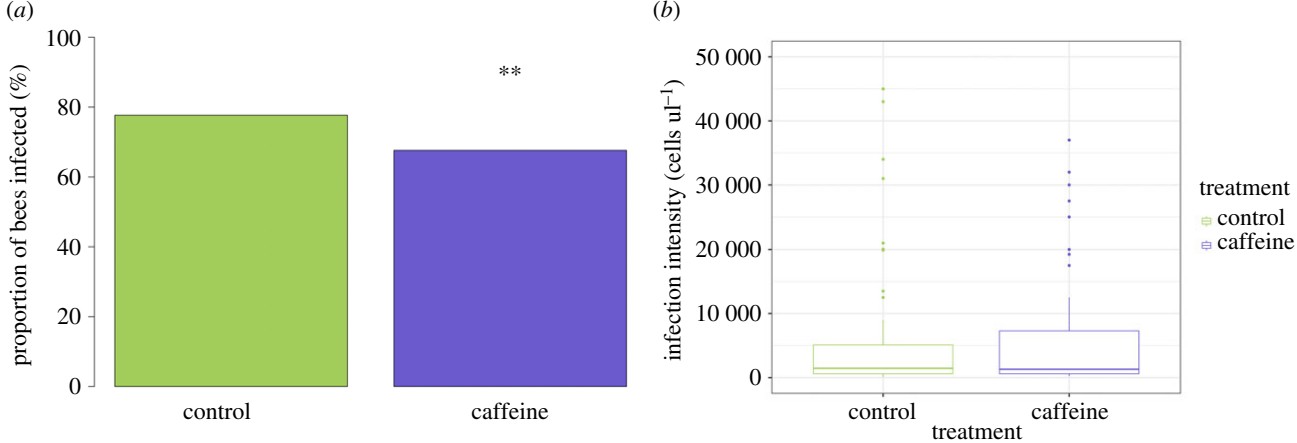

**Figure 5.** *Nosema bombi* infection prevalence (*a*) and infection intensity (*b*) for foraging bumblebees sampled during the experiment. There were significantly fewer bees that had *N. bombi* infections in the caffeine treatment when compared to the control group (significant difference marked with a double asterisk). By contrast, there was no significant difference in infection intensities between the caffeine and control groups. (Online version in colour.)

parasitaemia. Exposure to such compounds depends upon the duration of flowering and density of flowers, both of which can be high in AES schemes [20]. Caffeine is well known for its biological activity [56,57,61,62], and its impacts on *N. bombi* infection may be through reducing spore germination and subsequent infection success. This is likely to be enhanced in bumblebee larvae, whose blind gut may lead to an increase in caffeine concentration during the exposure period, resulting in a greater impact on the parasite and contributing to the reduction of infection prevalence and intensity recorded in adult bumblebees following eclosure. One alternative explanation is that caffeine consumption may be stimulating immune gene expression by imposing positive effects on carbohydrate metabolism pathways, as has been recorded in honeybees that were experimentally fed caffeine [56], and that this enhanced immunity may reduce infection success. Irrespective of the mechanism driving our results, understanding how caffeine may interact with other phytochemicals in pollen and nectar and how this in turn impacts pollinator disease epidemiology is critical in understanding the value of plant-based disease management strategies. Given that *in vitro* and *in vivo* work has identified synergistic effects of compounds on bumblebee pollinators [63,64], and that compounds and concentrations vary geographically, it is likely that schemes enhancing floral abundance and diversity will result in pollinators being exposed to a range of phytochemicals, some of which may even have negative effects [65].

These impacts of caffeine on individual infections were magnified at the colony level, with treated colonies having significantly lower prevalence and intensity of infections. In control colonies, the prevalence and intensity of infections increased across the colony cycle including the sexual production phase, which has implications for environmental disease persistence [66]. By contrast, both prevalence and intensity remained at relatively lower levels in treated colonies. This will have reduced the force of intracolonial infection, by lowering the number of infective spores in the social environment. It is also likely to have significant knock-on effects for transmission between colonies. As *N. bombi* is not species-specific in *Bombus* [67], the reduction in disease prevalence, recorded here, from caffeine consumption in foraging bumblebees may also reduce the likelihood of

interspecific disease transmission [68,69] and thus further impact environmental disease epidemiology. We would note that intracolonial transmission may have been facilitated in our foraging arenas as these may have allowed faeces to accumulate. However, even with this potential interaction, the prevalence was still significantly lower in our caffeine-treated colonies. Interestingly, the higher infection intensities we observed in earlier cohorts in control colonies were sufficient to yield higher colony infection prevalence. We suggest that this early *N. bombi* bloom in young adults, which caffeine inhibited, either through larval or adult consumption, as both may impact transgenerational disease in bumblebees [36], is critical in maintaining high intracolonial disease prevalence. Consequently, our results suggest that beneficial forage plants, such as those that produce caffeine, if provided throughout the lifecycle of a bumblebee colony (either by overlapping or prolonged anthesis periods) may mitigate the impact of the endemic and emergent disease. Moreover, it is important to note that the impact of the *N. bombi* bloom in young adult bumblebees on prevalence is likely to be greater in incipient colonies where there are fewer individuals. Consequently, younger bees will be involved in nest-based tasks including nursing behaviours [70], where they are likely to pass spores to developing larvae [71], thus contributing to disease amplification. As such, planting strategies, such as AES or the North American equivalent CRP, that target incipient colonies may have a greater impact on pollinator parasite epidemiology. The emergence of *N. bombi* in North America has been implicated as a driver of indigenous bumblebee declines [24]. Sainfoin is grown in North America and caffeine is present in the nectar of other plants found throughout North America [57]. Consequently, we recommend our results be studied in the context of North American bumblebee declines in response to EIDs. Our findings suggest that the incidence and impact of *N. bombi* can be mitigated with adaptive planting strategies, which include forage plants containing caffeine. Alongside the management of endemic disease, this approach may be implemented to provide a mechanism with which to minimize the spread of emergent disease to novel areas. Finally, our results suggest that the CRP, which has a broadly similar approach to AES, could be tailored to combat emerging pollinator disease, as *N. bombi* is currently

considered an emerging disease in North American bumblebee populations.

Infection with *N. bombi* dramatically impacts bumblebee colony health and fitness [72,73], and this may have implications for population persistence [24]. Phytochemicals can negatively impact bumblebee fitness, for example, by reducing the production of sexual castes [65]. By contrast, our results show that chronic caffeine consumption did not have a negative effect on the health or fitness of our experimental colonies; indeed caffeine-treated colonies had a trend for longer persistence and the production of more individuals of sexual castes, in contrast with what would typically be expected in *N. bombi* infected colonies [73]. Consequently, at the colony level, caffeine is not having a detrimental effect on brood development or on queen fecundity. Caffeine has been reported in 13 orders of plants [57]. Consequently, our findings on the impact of caffeine on bumblebee parasite epidemiology may be relevant in other landscapes, globally, where AES and sainfoin are not present. In addition, caffeine has been shown to enhance a pollinator's memory of reward [56], resulting in increased visitation rates by altering the foraging behaviour of bees. Our results suggest that such manipulation of pollinator behaviour by caffeine, resulting in repeated flower visitations [74], may also indirectly benefit foraging bumblebees by reducing the incidence and distribution of a key parasite, *N. bombi*, within a given environment. However, it should be noted that under field conditions caffeine consumption may lead to suboptimal foraging strategies, as caffeine-producing plants have been linked to an overestimation of forage quality in honeybees [75], which, if replicated in bumblebees, may have a negative impact on colony fitness through nutritional deficiencies.

The current guidelines and floral recommendations for AES have been developed with the scientific community [76,77], and AES have been shown to increase bumblebee species richness [20,21] and more recently bumblebee reproductive fitness [23]. By integrating multidisciplinary approaches which combine chemical and epidemiological studies for example, such as the results presented here, into AES, and similar schemes across the globe, such as CRP, their management could be further adapted to include species that have beneficial floral chemistry, which may provide indirect fitness benefits to pollinators through disease management.

## 4. Methodology

### (a) Phytochemical identification

Pollen and nectar were individually collected from nine plants included in AES that were described as potential bumblebee forage (electronic supplementary material, table S1). Chemical analysis was undertaken using LC-MS, where 41 and 27 phytochemicals (inclusive of isomers) were identified, respectively, from pollen and nectar sampled from the nine AES species (electronic supplementary material, tables S2 and S3). Caffeine was identified in the nectar of sainfoin (*Onobrychis viciifolia*) with Rt of 6.11 and a pseudomolecular ion with mass-to-charge ratio (*m/z*) of 195.09 and by comparison to a caffeine standard. In addition, caffeine was identified in sainfoin nectar using high-resolution mass spectrometry.

### (b) *Nosema bombi* inoculum

To elucidate the effect of caffeine on *N. bombi*, we used a larval inoculation paradigm, as larvae are the most susceptible stage to infection [78]. A wild *B. terrestris* queen that was naturally infected with *N. bombi* was caught from Windsor Great Park, UK (SU992703) in 2016 [36]. The infected queen's gut was isolated by dissection and homogenized in 0.01 M $NH_4Cl$. The resulting spore solution was centrifuged at 4°C for 10 min at 5000 rpm to isolate and purify the spore pellet. The pellet was resuspended in 100 µl of 0.01 M $NH_4Cl$ and the inoculum was checked for the presence of non-target bumblebee parasites using a compound phase-contrast microscope set to ×400 magnification. A larval *N. bombi* inoculant was prepared by combining inverted sugar water and pollen (3:1) to create an artificial worker feed as outlined in [71]. This was then combined in equal proportions (100 µl:100 µl) with the *N. bombi* inoculum to create an experimental inoculant which contained 50 000 spores/µl. For all experimental procedures caffeine was administered to bumblebees via sugar water, caffeine was not administered to bees using pollen. All pollen, which came from mixed flowers (Biobest, Belgium), used in all experimental procedures was irradiated to kill any microbes and a sample of each pollen consignment was visually inspected to check for bumblebee parasites using a compound phase-contrast microscope set to ×400 magnification. While pollen was not screened for phytochemicals, it was provided to both control and experimental treatments across all experiments. Consequently, as pollen was from mixed flowers, our methodology mimicked natural intake of pollen into bumblebee colonies [79] with the experimental addition of caffeine to sugar water representing the addition of sainfoin to this intake.

### (c) Investigating the impact of caffeine on *Nosema bombi* infection on individual epidemiology

Eight *B. terrestris audax* colonies (hereafter referred to as donor colonies) containing a queen, brood and a mean of 45 (±6.5 s.e.) workers were obtained from Biobest, Belgium. Colonies were kept in a dark room at 26°C and 50% humidity (red light was used for any colony manipulation). To ensure colonies were healthy and developing normally they were monitored for 7 days prior to use in any experimental procedures. This included randomly screening 10% of the workers every 2 days, from each colony, for common parasitic infections (*Apicystis bombi*, *Crithidia/bombi* and *N. bombi*) in faeces using a phase-contrast microscope set to ×400 magnification. No infections were identified in any of the eight donor colonies.

Micro-colonies were established by removing eight patches of brood containing approximately 10 developing larvae (growth stage L2–3), from each of the eight donor colonies. Each of these patches of brood was placed in an individual $140 \times 80 \times 55$ mm acrylic box. These micro-colonies were each provisioned with *ad libitum* pollen and sugar water, and three workers from their original donor colony to provide brood care. Prior to being entered into the experiment all brood-caring workers were individually marked using a coloured, numbered Opalith tag and recorded so that they could be distinguished from newly eclosed workers.

To investigate if caffeine had any prophylactic properties, 16 micro-colonies (2 per donor colony) as described above were used. Prior to inoculation, control larvae were kept in their original micro-colonies ($n = 8$) and provided *ad libitum* pollen and sugar water. However, in the experimental groups, *ad libitum* pollen and sugar water containing caffeine (Sigma Aldrich CO750) at 200 µM ($n = 8$ micro-colonies) were provided for 7 days. Caffeine was added to sugar water using 4 ml of 40% $MeOH\,l^{-1}$ as a solvent, control colonies also had 4 ml of 40%

MeOH l$^{-1}$ added. After 7 days, both experimental and control larvae at either instar 2 or 3 were artificially inoculated with 50 000 *N. bombi* spores in 4.3 µl of inoculant (described above) using a 20 µl pipette. The larvae were left to consume the inoculum for 30 min, before being returned to their micro-colony.

In a simultaneous experiment, the therapeutic effect of caffeine was investigated. Here, 16 micro-colonies (2 per donor colony) as described above were used. In contrast with the prophylactic investigation, larvae in the therapeutic investigation were each inoculated with 50 000 *N. bombi* spores in 4.3 µl of experimental inoculant, as described above, using a 20 µl pipette, prior to experimental feeding. The inoculated larvae were returned to their respective micro-colonies with brood-caring workers. Each control micro-colony ($n = 8$) was provisioned with *ad libitum* pollen and sugar water. However, in the experimental groups, *ad libitum* pollen and sugar water containing caffeine at 200 µM ($n = 8$ micro-colonies) were provided for 7 days. As before, phytochemicals were added to sugar water using 4 ml of 40% MeOH l$^{-1}$; control colonies also had 4 ml of 40% MeOH l$^{-1}$ of sugar water.

In both feeding trials, larvae were allowed to develop naturally and pupate in their respective micro-colonies. Once eclosed, new workers were marked using a coloured, numbered Opalith tag and individually quarantined for 3 days in an inverted plastic cup, which was modified with a hole that enabled a 15 ml falcon tube to be inserted. The falcon tube contained control inverted sugar water diluted with double-distilled H$_2$O (50% w/w) that the newly eclosed workers could feed on. At the end of the quarantine period, each worker had its thorax width measured (mm) as a proxy for bumblebee size, using a set of Mitutoyo digital calipers. In addition, each worker was isolated in a 25 ml plastic vial where it provided a faecal sample, collected in a 10 µl glass capillary, the volume (µl) of which was recorded. Following this, each newly eclosed worker's faecal sample was screened for *N. bombi* by microscopic examination using a phase-contrast microscope at ×400 magnification. If an infection was identified a Neubauer improved haemocytometer was used to quantify the parasite load. Workers were then sacrificed and stored in a labelled Eppendorf tube at −80°C.

## (d) Investigating the impact of caffeine on the social epidemiology of *Nosema bombi*

Between February and April 2018, 250 wild, foraging *B. terrestris* queens were collected from Windsor Great Park, Surrey, UK (SU992703). All queens were screened for common bumblebee endoparasites (described above, including *Sphaerularia bombi* as this infects queen bumblebees) via microscopic examination of faeces using a phase-contrast microscope at × 400 magnification. Following this initial screen, apparently uninfected bumblebee queens were quarantined in an individual 127 × 67 × 50 mm acrylic box, where they were fed *ad libitum* pollen and sugar water, in a dedicated bumblebee rearing room, which was kept at 26°C and 50% humidity. Seven days after the initial parasite screen each queen was rescreened for the common bumblebee endoparasites as above. This seven-day delay ensured that any incipient infections missed during the initial screen were identified once parasitaemia had increased to detectable levels [80]. All queens that were infected during either of the parasite screens ($n = 15$) were excluded from the experiment and released back to the original field collection site. Uninfected queens (hereafter referred to as queens) were returned to their acrylic box in the dedicated rearing room (described above) and entered into the experiment. All queens were provided with *ad libitum* sugar water from a gravity feeder and a pollen ball for nutrition and to encourage egg-laying. Pollen balls were created by combining pollen and sugar water (50% w/w) in a 20 : 1 ratio. This mixture was then subdivided into individual 15 × 15 × 15 mm cubes.

Sugar water and pollen balls were changed every 7 days or if a queen had spoiled her resources, whichever came first. Queens were left to develop naturally, and reproductive output was monitored and recorded. Once the first set of workers had hatched, these were marked using numbered, coloured Opalith tags (tag colour was unique to cohort, not individual or colony), and the remaining incipient brood was inoculated with *N. bombi* (as described above) and entered into one of two feeding regimes. Any queen that did not lay a clutch of eggs within eight weeks ($n = 113$) was excluded from the experiment and released back to the original field collection site. Of the 250 queens originally caught, 40 produced an incipient colony, of which 19 lasted for the duration of the experiment.

Following inoculation, each incipient colony was placed into a 290 × 220 × 130 mm plastic colony box. This habitation box was connected, via a plastic tube, to a separate 290 × 220 × 130 mm plastic box that was used as a foraging arena. The foraging arena provided *ad libitum* pollen and experimental sugar water (as described above). Each control colony ($n = 10$ colonies) was provisioned with *ad libitum* pollen and sugar water for the duration of the colony lifecycle. In the experimental group *ad libitum* pollen and sugar water, containing caffeine at 200 µM ($n = 9$ colonies) was provided for the duration of the colony life cycle. Pollen and sugar water were changed every 7 days, or once the colony had consumed all of the resources, whichever came first.

To investigate if caffeine had any effect on *N. bombi* epidemiology, a range of colony specific measurements were taken throughout the colony lifecycle. Every two weeks, all newly eclosed workers were individually marked using a coloured, high varnish paint, which was unique to that specific brood cohort and not the colony. To investigate the impact of caffeine on infection prevalence and intensity in foragers, which are the route for inter-colony transmission of the parasite, every 7 days 10% of the total colony population that were foraging were removed and screened for *N. bombi* infection via microscopic examination of the faeces. If an infection was identified, a Neubauer improved haemocytometer was used to calculate the infection intensity (cells per µl). Finally, when the colony was terminated all remaining bees were screened for *N. bombi* infection as described above. To ensure consistency across colonies with respect to termination, the following parameters were used to define the end of a colony. Colony endpoint was defined as either three weeks following the death of the original queen, three weeks following the eclosure of the first sexual caste or three weeks after the queen last laid a clutch of eggs. These guidelines ensured that all developing brood would reach eclosure by the endpoint, enabling a robust estimation of the complete reproductive output of a colony.

## (e) Statistical analysis

All statistical analyses and graphical outputs were undertaken in R open-source programming language [81,82]. Chi-squared ($\chi^2$) tests were used to compare the infection prevalence between treatments in both the therapeutic and prophylactic investigations. To analyse the therapeutic and prophylactic effect of caffeine on *N. bombi* infection intensity in newly eclosed workers, and the effect of caffeine on colony-level prevalence, infection intensity and dynamics LMM models were constructed in the R package 'lme4' [83] (electronic supplementary material, methods).

Data accessibility. Data have been made open access and deposited onto FigShare under the title 'Agri-environment scheme nectar chemistry negatively impacts bumblebee parasite epidemiology' https://doi.org/10.6084/m9.figshare.13668767.

The data are provided in the electronic supplementary material [84].

**Authors' contributions.** A.J.F.: conceptualization, data curation, formal analysis, investigation, methodology, project administration, writing-original draft, writing-review and editing; H.K.: investigation, methodology, writing-review and editing; I.W.F.: data curation, formal analysis, methodology, writing-review and editing; P.C.S.: conceptualization, funding acquisition, methodology, project administration, resources, supervision, writing-review and editing; M.JF.B.: conceptualization, funding acquisition, methodology, project administration, resources, supervision, writing-review and editing.

All authors gave final approval for publication and agreed to be held accountable for the work performed therein.

**Competing interests.** We declare we have no competing interests.

**Funding.** This work was funded by a Biotechnology and Biological Sciences Research Council Doctoral Training Program Studentship (DTP1 BB/J014575/1) and the Peter Sowerby Foundation funded by the Ann Sowerby Post-doctoral Fellowship in Pollinator Health at Royal Botanic Gardens, Kew.

**Acknowledgements.** We would like to thank Lucy Thursfield and Sue Baldwin for technical support and Harry Siviter for statistical advice. We would also like to thank Emorsgate seeds for allowing us to sample flowers on their land, and Windsor Great Park for allowing us to collect wild bumblebees. A.J.F. would also like to thank Isla for enriching his life.

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
