## [Peer Review File · Proceedings of the Royal Society B: Biological Sciences]

Review History

RSPB-2021-0363.R0 (Original submission)

Review form: Reviewer 1

Recommendation

Accept with minor revision (please list in comments)

Scientific importance: Is the manuscript an original and important contribution to its field?

Good

General interest: Is the paper of sufficient general interest?

Good

Quality of the paper: Is the overall quality of the paper suitable?

Acceptable

Is the length of the paper justified?

No

Should the paper be seen by a specialist statistical reviewer?

No

Do you have any concerns about statistical analyses in this paper? If so, please specify them explicitly in your report.

No

It is a condition of publication that authors make their supporting data, code and materials available - either as supplementary material or hosted in an external repository. Please rate, if applicable, the supporting data on the following criteria.

Is it accessible?

Yes

Is it clear?

Yes

Is it adequate?

Yes

Do you have any ethical concerns with this paper?

No

Comments to the Author

The authors report an interesting and important study documenting the importance of phytochemicals in protecting pollinators from pathogens.

The study is well designed and the results clearly support the claims by the author on the potential benefits of caffeine.

Caffeine as a beneficial phytochemical has been recently researched in the context of pollinator health with specific reference to honeybees and their pathogen *Nosema ceranae*. Given that the authors here document benefits of caffeine on a different species of *Nosema*, it strengthens their case to cite these other studies. In addition, phytochemicals benefiting pollinator health is also well documented and authors here fail to cite some important recent publications.

The discussion section is rather weak as the authors do not propose any suggestions for mode of action of caffeine but instead just repeat the introduction premise on the importance of pollinator habitat plants.

The study is fairly small but the results are clear.

Please cite the following publications during revision and improve discussion to include some of the ideas proposed in these publications

Lu, Y.-H., et al. (2020). "Identification of Immune Regulatory Genes in *Apis mellifera* through Caffeine Treatment." *Insects* 11(8): 516.

Bernklau, E., et al. (2019). "Dietary Phytochemicals, Honey Bee Longevity and Pathogen Tolerance." *Insects* 10(1): 14.

Palmer-Young, E. C., et al. (2017). "Synergistic effects of floral phytochemicals against a bumble bee parasite." *Ecology and Evolution* 7(6): 1836-1849.

Masai Biller, L. A., Rebecca Irwin, Caitlin McAllister, Evan Palmer-Young (2015). "Possible Synergistic Effects of Thymol and Nicotine against *Crithidia bombi* Parasitism in Bumble Bees." *PLoS ONE* 10(12): e0144668.

Review form: Reviewer 2

Recommendation

Accept with minor revision (please list in comments)

Scientific importance: Is the manuscript an original and important contribution to its field?

Excellent

General interest: Is the paper of sufficient general interest?

Excellent

Quality of the paper: Is the overall quality of the paper suitable?

Good

Is the length of the paper justified?

Yes

Should the paper be seen by a specialist statistical reviewer?

No

Do you have any concerns about statistical analyses in this paper? If so, please specify them explicitly in your report.

No

It is a condition of publication that authors make their supporting data, code and materials available - either as supplementary material or hosted in an external repository. Please rate, if applicable, the supporting data on the following criteria.

Is it accessible?

Yes

Is it clear?

Yes

Is it adequate?

No

Do you have any ethical concerns with this paper?

No

Comments to the Author

The authors Folly et al, explore the phytochemicals present in the nectar and pollen of plants planted as part of Agri-environmental schemes (AES) and determine if caffeine, a chemical found in one of the screened plants (Sainfoin), can have a beneficial effect on bumblebees exposed to the parasite *Nosema bombi*. Manipulative experiments examined the prophylactic and therapeutic effects of caffeine on queen less microcolonies established from commercially sourced *Bombus terrestris* colonies and exposed to *Nosema bombi*. In addition, whole colony effects were explored using wild caught queen *B.terrestris* queens. The results show that caffeine can have both a prophylactic and therapeutic effect on *N. bombi* infection and in whole colony caffeine can reduce infection intensity particularly in younger bumblebees.

Overall, this was an enjoyable paper to read with well laid out justification for the experimental work undertaken with clearly outlined methods and analysis employed and clearly presented results.

I have a concern that the paper may sometimes over inflate the link between AES and their findings. Whilst for example I agree with statements on lines 215-216 "AES and CRP may have unexplored benefits as they have the potential to be co-opted as disease management tools to further improve pollinator health in the field" , this study's findings relate only to one phytochemical and it is unknown whether the natural cocktail of chemicals will have the same net effect. It is for that reason, I also feel the title and general tone of the manuscript is too broad-brush and should be toned down.

Introduction

The introduction is succinct and informative, however, the relevance of *N. bombi* to bumblebee populations, which is not mentioned until its brief appearance in the discussion could be expanded upon to help provide better context for the importance of the study.

Methods

The methods should better integrate the chemistry work. These are currently as supplementary methods but there is no clear link to them in the main manuscript. The manuscript would benefit from at least an overview of what has been done included in the methods and reference to the supplementary methods.

Results

The 'Phytochemical identification' column, which is differently named in supplementary table 3, despite referring to the same concept, should be renamed to 'Phytochemical identified' for clarity. Additionally, abbreviations are used in both supplementary table 2 and supplementary table 3 without being included in the table legends. In Figure 1, poor resolution and small size make reading the y-label difficult. Additionally, it is not clear why in line 135 there is reference to Figure 3. For Figure 3, the in-figure legend is unnecessary. In line 417, the word 'number' is misspelled.

Discussion

Throughout the report it does not become clear what the relevance of CRP is to the manuscript, as to my understanding, only plants involved in the AES scheme are included. Is it only included because of its similar approach to AES? Clarification needed. While the discussion briefly mentions field applicability, a broader exploration of this could be beneficial. This could involve a brief mention of potential interactions with other phytochemicals, exposure to different doses of caffeine or a further exploration of what is meant by increased potential for nutritional deficiencies. While it is clear that in the experiment food is freely provided, which would not be the case for field conditions, it is not clear why the presence of flowers which have the alkaloid would be more likely to lead to nutritional deficiencies. Perhaps, if such flowers are less nutritious themselves or through influencing foraging choices? In any clarity on this could improve the overall understanding of the topic for the reader.

Line 52: font size changes

Line 94: may benefit from some system specific references

217-232: what about possible negative effects of phytochems?

254: clarify its not species specific within *Bombus*.

271-272: are the planting schemes in the US?

307: what was the concentration of the inoculant? How was the inoculant also deemed clear of other parasites?

308-309: mention the use of UV irradiation to remove microbes, a necessary step for the experiment to ensure that the bees are only inoculated with the pathogen being assessed in the study. However, the sterilisation of different bee pathogens require different doses of irradiation - is it possible to include details on this or the supplier of the pollen.

309: was the pollen of a known flower? Was it tested for phytochemicals?

315: nice to see you checked repeatedly for adult parasite presence

329: to clarify, was caffeine placed on pollen at all?

330: though you describe the volume of MeOH added to the sucrose, you haven't noted the volume of sugar water to interpret what the final concentration of the solvent was.

350: For clarity, which workers were isolated to provide faecal samples - the recently eclosed workers or the original 'nurse' workers?

361-366: good practice to isolate and recheck health like this

379-380: those pesky queens!

376: it is not clear if the number of brood inoculated was controlled/standardised between colonies, and if not, was to total number of spores applied to the colony standardised? also was it just the L2/3 brood inoculated?

383: was there cat-litter or similar in the base to absorb faeces or could pools of faeces in the arena facilitate dispersal between foraging workers?

386: Did this caffeine treatment include MeOH as before?

431-432: It would be nice to include these normality plots as supplementary

Some minor formatting inconsistencies;

In line 520: '10' is placed in bold which is inconsistent with the rest of the formatting.

Lines 575-576: a grey background is included, which is inconsistent with other references.

Decision letter (RSPB-2021-0363.R0)

09-Apr-2021

Dear Dr Folly:

Your manuscript has now been peer reviewed and the reviews have been assessed by an Associate Editor. The reviewers' comments (not including confidential comments to the Editor) and the comments from the Associate Editor are included at the end of this email for your reference. As you will see, the reviewers and the Associate Editor have raised some concerns with your manuscript and we would like to invite you to revise your manuscript to address them.

Research ethics:

Use of animals and field studies:

If your study uses animals please include details in the methods section of any approval and licences given to carry out the study and include full details of how animal welfare standards

were ensured. Field studies should be conducted in accordance with local legislation; please include details of the appropriate permission and licences that you obtained to carry out the field work.

It is a condition of publication that you make available the data and research materials supporting the results in the article. Please see our Data Sharing Policies (<https://royalsociety.org/journals/authors/author-guidelines/#data>). Datasets should be deposited in an appropriate publicly available repository and details of the associated accession number, link or DOI to the datasets must be included in the Data Accessibility section of the article (<https://royalsociety.org/journals/ethics-policies/data-sharing-mining/>). Reference(s) to datasets should also be included in the reference list of the article with DOIs (where available).

Please submit a copy of your revised paper within three weeks. If we do not hear from you within this time your manuscript will be rejected. If you are unable to meet this deadline please let us know as soon as possible, as we may be able to grant a short extension.

Best wishes,
Professor Hans Heesterbeek
<mailto:proceedingsb@royalsociety.org>

Associate Editor
Board Member: 1

Comments to Author:

Both reviewers are convinced by the importance of the study and the strength of the experimental design. Both also ask for some of the arguments in the discussion, e.g. the importance to CRP,

extrapolating from the study of "only" one compound, to either be clarified or turned down a bit. I agree with both sentiments. Because the requested changes are editorial and the reviews comments give detailed guidance, I would recommend accepting the manuscripts, but urge the authors to carefully consider the reviewers comments.

Reviewer(s)' Comments to Author:

Referee: 1

Comments to the Author(s)

The authors report an interesting and important study documenting the importance of phytochemicals in protecting pollinators from pathogens.

The study is well designed and the results clearly support the claims by the author on the potential benefits of caffeine.

Caffeine as a beneficial phytochemical has been recently researched in the context of pollinator health with specific reference to honeybees and their pathogen *Nosema ceranae*. Given that the authors here document benefits of caffeine on a different species of *Nosema*, it strengthens their case to cite these other studies. In addition, phytochemicals benefiting pollinator health is also well documented and authors here fail to cite some important recent publications.

The discussion section is rather weak as the authors do not propose any suggestions for mode of action of caffeine but instead just repeat the introduction premise on the importance of pollinator habitat plants.

The study is fairly small but the results are clear.

Please cite the following publications during revision and improve discussion to include some of the ideas proposed in these publications

Lu, Y.-H., et al. (2020). "Identification of Immune Regulatory Genes in *Apis mellifera* through Caffeine Treatment." *Insects* 11(8): 516.

Bernklau, E., et al. (2019). "Dietary Phytochemicals, Honey Bee Longevity and Pathogen Tolerance." *Insects* 10(1): 14.

Palmer-Young, E. C., et al. (2017). "Synergistic effects of floral phytochemicals against a bumble bee parasite." *Ecology and Evolution* 7(6): 1836-1849.

Masai Biller, L. A., Rebecca Irwin, Caitlin McAllister, Evan Palmer-Young (2015). "Possible Synergistic Effects of Thymol and Nicotine against *Crithidia bombi* Parasitism in Bumble Bees." *PLoS ONE* 10(12): e0144668.

Referee: 2

Comments to the Author(s)

The authors Folly et al, explore the phytochemicals present in the nectar and pollen of plants planted as part of Agri-environmental schemes (AES) and determine if caffeine, a chemical found in one of the screened plants (Sainfoin), can have a beneficial effect on bumblebees exposed to the parasite *Nosema bombi*. Manipulative experiments examined the prophylactic and therapeutic effects of caffeine on queen less microcolonies established from commercially sourced *Bombus terrestris* colonies and exposed to *Nosema bombi*. In addition, whole colony effects were explored using wild caught queen *B. terrestris* queens. The results show that caffeine can have both a prophylactic and therapeutic effect on *N. bombi* infection and in whole colony caffeine can reduce infection intensity particularly in younger bumblebees.

Overall, this was an enjoyable paper to read with well laid out justification for the experimental work undertaken with clearly outlined methods and analysis employed and clearly presented results.

I have a concern that the paper may sometimes over inflate the link between AES and their findings. Whilst for example I agree with statements on lines 215-216 "AES and CRP may have unexplored benefits as they have the potential to be co-opted as disease management tools to further improve pollinator health in the field" , this study's findings relate only to one phytochemical and it is unknown whether the natural cocktail of chemicals will have the same

net effect. It is for that reason, I also feel the title and general tone of the manuscript is too broad-brush and should be toned down.

Introduction

The introduction is succinct and informative, however, the relevance of *N. bombi* to bumblebee populations, which is not mentioned until its brief appearance in the discussion could be expanded upon to help provide better context for the importance of the study.

Methods

The methods should better integrate the chemistry work. These are currently as supplementary methods but there is no clear link to them in the main manuscript. The manuscript would benefit from at least an overview of what has been done included in the methods and reference to the supplementary methods.

Results

The 'Phytochemical identification' column, which is differently named in supplementary table 3, despite referring to the same concept, should be renamed to 'Phytochemical identified' for clarity. Additionally, abbreviations are used in both supplementary table 2 and supplementary table 3 without being included in the table legends. In Figure 1, poor resolution and small size make reading the y-label difficult. Additionally, it is not clear why in line 135 there is reference to Figure 3. For Figure 3, the in-figure legend is unnecessary. In line 417, the word 'number' is misspelled.

Discussion

Throughout the report it does not become clear what the relevance of CRP is to the manuscript, as to my understanding, only plants involved in the AES scheme are included. Is it only included because of its similar approach to AES? Clarification needed. While the discussion briefly mentions field applicability, a broader exploration of this could be beneficial. This could involve a brief mention of potential interactions with other phytochemicals, exposure to different doses of caffeine or a further exploration of what is meant by increased potential for nutritional deficiencies. While it is clear that in the experiment food is freely provided, which would not be the case for field conditions, it is not clear why the presence of flowers which have the alkaloid would be more likely to lead to nutritional deficiencies. Perhaps, if such flowers are less nutritious themselves or through influencing foraging choices? In any clarity on this could improve the overall understanding of the topic for the reader.

Line 52: font size changes

Line 94: may benefit from some system specific references

217-232: what about possible negative effects of phytochemicals?

254: clarify its not species specific within *Bombus*.

271-272: are the planting schemes in the US?

307: what was the concentration of the inoculant? How was the inoculant also deemed clear of other parasites?

308-309: mention the use of UV irradiation to remove microbes, a necessary step for the experiment to ensure that the bees are only inoculated with the pathogen being assessed in the study. However, the sterilisation of different bee pathogens require different doses of irradiation - is it possible to include details on this or the supplier of the pollen.

309: was the pollen of a known flower? Was it tested for phytochemicals?

315: nice to see you checked repeatedly for adult parasite presence

329: to clarify, was caffeine placed on pollen at all?

330: though you describe the volume of MeOH added to the sucrose, you haven't noted the volume of sugar water to interpret what the final concentration of the solvent was.

350: For clarity, which workers were isolated to provide faecal samples - the recently eclosed workers or the original 'nurse' workers?

361-366: good practice to isolate and recheck health like this

379-380: those pesky queens!

376: it is not clear if the number of brood inoculated was controlled/standardised between colonies, and if not, was to total number of spores applied to the colony standardised? also was it just the L2/3 brood inoculated?

383: was there cat-litter or similar in the base to absorb faeces or could pools of faeces in the arena facilitate dispersal between foraging workers?

386: Did this caffeine treatment include MeOH as before?

431-432: It would be nice to include these normality plots as supplementary

Some minor formatting inconsistencies;

In line 520: '10' is placed in bold which is inconsistent with the rest of the formatting.

Lines 575-576: a grey background is included, which is inconsistent with other references.

Author's Response to Decision Letter for (RSPB-2021-0363.R0)

See Appendix A.

Decision letter (RSPB-2021-0363.R1)

05-May-2021

Dear Dr Folly

I am pleased to inform you that your manuscript entitled "Agri-environment scheme nectar chemistry can suppress the social epidemiology of parasites in an important pollinator" has been accepted for publication in Proceedings B.

Data Accessibility section

Open Access

Paper charges

Sincerely,

Professor Hans Heesterbeek

Associate Editor:

Board Member

Comments to Author:

The authors replied and adopted reviewers comments comprehensively and I am happy to recommend the revised version of the manuscript for publication.

Appendix A

Folly et al 2021 PRSB responses to reviewers:

Associate Editor Board Member: 1:

Both reviewers are convinced by the importance of the study and the strength of the experimental design. Both also ask for some of the arguments in the discussion, e.g. the importance to CRP, extrapolating from the study of "only" one compound, to either be clarified or turned down a bit. I agree with both sentiments. Because the requested changes are editorial and the reviews comments give detailed guidance, I would recommend accepting the manuscripts, but urge the authors to carefully consider the reviewers comments.

Authors: We would like to thank the associate board member and both reviewers for taking the time to review our manuscript. We are especially pleased that both reviewers are convinced by the importance of the study and the strength of the experimental design and to the associate editor for recommending acceptance of the manuscript. We address specific concerns raised by the reviewers below.

Referee 1: *The authors report an interesting and important study documenting the importance of phytochemicals in protecting pollinators from pathogens.*

The study is well designed and the results clearly support the claims by the author on the potential benefits of caffeine.

Authors: We thank the reviewer for noting both the interest and importance of our study and for their comments on the strength of our experimental design

Referee 1: *Caffeine as a beneficial phytochemical has been recently researched in the context of pollinator health with specific reference to honeybees and their pathogen *Nosema ceranae*. Given that the authors here document benefits of caffeine on a different species of *Nosema*, it strengthens their case to cite these other studies. In addition, phytochemicals benefiting pollinator health is also well documented and authors here fail to cite some important recent publications.*

The discussion section is rather weak as the authors do not propose any suggestions for mode of action of caffeine but instead just repeat the introduction premise on the importance of pollinator habitat plants.

The study is fairly small but the results are clear.

Authors: We agree with the reviewer and have included the references they have suggested to strengthen our findings and provide wider context. We have also enhanced the strength of our discussion section by including the following lines:

L230-231: which can suppress latent virus infection and reduce *N. ceranae* spore intensity in honeybees [56, 57].

L248-258: One alternative explanation is that caffeine consumption may be stimulating immune gene expression by imposing positive effects on carbohydrate metabolism pathways, as has been recorded in honeybees who were experimentally fed caffeine [56], and that this enhanced immunity may reduce infection success. Irrespective of the mechanism driving our results, understanding how caffeine may interact with other phytochemicals in pollen and nectar and how this in turn impacts pollinator disease epidemiology is critical in understanding the value of plant-based disease management strategies. Given that *in vitro* and *in vivo* work has identified synergistic effects of compounds on bumblebee pollinators [63, 64], and that compounds and concentrations vary geographically, it is likely that schemes that enhance floral abundance and diversity will result in pollinators being exposed to a range of phytochemicals, some of which may even have negative effects [65].

Referee 1: *Please cite the following publications during revision and improve discussion to include some of the ideas proposed in these publications*

*Lu, Y.-H., et al. (2020). "Identification of Immune Regulatory Genes in *Apis mellifera* through Caffeine Treatment." *Insects* 11(8): 516.*

*Bernklau, E., et al. (2019). "Dietary Phytochemicals, Honey Bee Longevity and Pathogen Tolerance." *Insects* 10(1): 14.*

Palmer-Young, E. C., et al. (2017). "Synergistic effects of floral phytochemicals against a bumble bee parasite." *Ecology and Evolution* 7(6): 1836-1849.

Masai Biller, L. A., Rebecca Irwin, Caitlin McAllister, Evan Palmer-Young (2015). "Possible Synergistic Effects of Thymol and Nicotine against *Crithidia bombi* Parasitism in Bumble Bees." *PLoS ONE* 10(12): e0144668.

Authors: As per the comment above, above these references have now been included and we thank the reviewer for suggesting them, as they clearly enhance our manuscript.

Referee 2: *The authors Folly et al, explore the phytochemicals present in the nectar and pollen of plants planted as part of Agri-environmental schemes (AES) and determine if caffeine, a chemical found in one of the screened plants (Sainfoin), can have a beneficial effect on bumblebees exposed to the parasite Nosema bombi. Manipulative experiments examined the prophylactic and therapeutic effects of caffeine on queen less microcolonies established from commercially sourced Bombus terrestris colonies and exposed to Nosema bombi. In addition, whole colony effects were explored using wild caught queen B.terrestris queens. The results show that caffeine can have both a prophylactic and therapeutic effect on N. bombi infection and in whole colony caffeine can reduce infection intensity particularly in younger bumblebees.*

Overall, this was an enjoyable paper to read with well laid out justification for the experimental work undertaken with clearly outlined methods and analysis employed and clearly presented results.

Authors: We thank the reviewer for their overview of our work and we are delighted that they enjoyed reading the manuscript.

Referee 2: *I have a concern that the paper may sometimes over inflate the link between AES and their findings. Whilst for example I agree with statements on lines 215-216 "AES and CRP may have unexplored benefits as they have the potential to be co-opted as disease management tools to further improve pollinator health in the field", this study's findings relate only to one phytochemical and it is unknown whether the natural cocktail of chemicals will have the same net effect. It is for that reason, I also feel the title and general tone of the manuscript is too broad-brush and should be toned down.*

Authors: We acknowledge that this study has investigated one bioactive compound recovered from AES plants. However, we actually provided evidence in the supplementary tables for the detection of a suite of compounds from AES plants. One of these, Biochanin A, has also been shown to have an impact on infection intensity in the bumblebee-Nosema system (Folly, A.J., et al 2020. Age related pharmacodynamics in a bumblebee-microsporidian system mirror similar patterns in vertebrates. *J Exp. Biol.* **223**:217828). We agree with the author that how combinations of phytochemicals may interact remains unclear. We have provided some context by including references which investigate synergistic effects of phytochemicals on bumblebee parasites (Palmer-Young, E. C., et al. (2017). Synergistic effects of floral phytochemicals against a bumble bee parasite. *Ecology and Evolution* **7**: 1836-1849 & Masai Biller et al. (2015). Possible Synergistic Effects of Thymol and Nicotine against *Crithidia bombi* Parasitism in Bumble Bees. *PLoS ONE* **10**: e0144668.) as suggested by reviewer 1, and we have also included the following text:

L248-258: However, understanding how caffeine may interact with other phytochemicals in nectar and how this in turn impacts pollinator disease epidemiology is critical in understanding the value of plant-based disease management strategies. Given that *in vitro* and *in vivo* work has identified synergistic effects of compounds on bumblebee pollinators [63, 64], and that compounds and concentrations vary geographically, it is likely that schemes that enhance floral abundance and diversity will result in pollinators being exposed to a range of phytochemicals, some of which may even have negative effects [65].

and edited text to provide a more precise description of our findings:

L210-212: Here we show that caffeine, which may be encountered by bumblebees in AES landscapes, can negatively impact both the individual and social epidemiology of a key bumblebee pathogen, *N. bombi*. Consumption of caffeine reduced the overall infection prevalence and intensity of *N. bombi* infections

We believe that throughout the rest of the manuscript it is clear that our results and the context we put them in relates to caffeine and not AES/CRP as a whole.

We have also changed the title to “**Agri-environment scheme nectar chemistry can suppress the social epidemiology of parasites in an important pollinator**” which we believe makes the statement less absolute, as requested by the reviewer

Referee 2: Introduction

*The introduction is succinct and informative, however, the relevance of *N. bombi* to bumblebee populations, which is not mentioned until its brief appearance in the discussion could be expanded upon to help provide better context for the importance of the study.*

Authors: We thank the reviewer for their description of our introduction. We actually did include text in the original Introduction about *Nosema bombi* (see below) and we hope that this addresses the concern of the reviewer

L61-62. One such emergent pathogen is the microsporidian *Nosema bombi* [44], which has been implicated in the population and range declines recorded in a suite of North American bumblebees

Referee 2: Methods

The methods should better integrate the chemistry work. These are currently as supplementary methods but there is no clear link to them in the main manuscript. The manuscript would benefit from at least an overview of what has been done included in the methods and reference to the supplementary methods.

Authors: We agree with the reviewer and have included a paragraph that broadly describes the chemistry work with a link to the supplementary methods:

L320-328: Phytochemical identification

Pollen and nectar were individually collected from nine plants included in AES that were described as potential bumblebee forage (Supplementary table 1). Chemical analysis was undertaken using LC-MS (Supplementary methods 1) where forty-one and twenty-seven phytochemicals (inclusive of isomers) were identified respectively from pollen and nectar sampled from the nine AES species (Supplementary table 2 & 3). Caffeine was identified in the nectar of sainfoin (*Onobrychis viciifolia*) with a retention time (rt) of 6.11 and a pseudomolecular ion with mass-to-charge ratio (m/z) of 195.09 and by comparison to a caffeine standard. In addition, caffeine was identified in sainfoin nectar using High Resolution Mass Spectrometry.

Referee 2: Results

The ‘Phytochemical identification’ column, which is differently named in supplementary table 3, despite referring to the same concept, should be renamed to ‘Phytochemical identified’ for clarity. Additionally, abbreviations are used in both supplementary table 2 and supplementary table 3 without being included in the table legends. In Figure 1, poor resolution and small size make reading the y-label difficult. Additionally, it is not clear why in line 135 there is reference to Figure 3. For Figure 3, the in-figure legend is unnecessary. In line 417, the word ‘number’ is misspelled.

Authors: We thank the reviewer for noting these points. We have updated the table as suggested (now in Folly et al 2021 PRSB Supplementary files). We have included updated images for figure 1 as a separate file with higher, publication ready resolution. We have removed the reference to figure 3 from line 135 and corrected the misspelling.

Referee 2: Discussion

Throughout the report it does not become clear what the relevance of CRP is to the manuscript, as to my understanding, only plants involved in the AES scheme are included. Is it only included because of its similar approach to AES? Clarification needed. While the discussion briefly mentions field applicability, a broader exploration of this could be beneficial. This could involve a brief mention of potential interactions with other phytochemicals, exposure to different doses of caffeine or a further exploration of what is meant by increased potential for nutritional deficiencies. While it is clear that in the experiment food is freely provided, which would not be the case for field conditions, it is not clear why the presence of flowers which have the alkaloid would be more likely to lead to nutritional deficiencies. Perhaps, if such flowers are less nutritious themselves or through influencing foraging choices? In any clarity on this could improve the overall understanding of the topic for the reader.

Authors: We thank the reviewer for these helpful comments and can confirm that comparisons with CRP are due to similarities within the schemes and provide a broader global view of how planting strategies could be tailored to combat endemic and emerging disease. Critically, *N. bombi* has been described as an emerging disease in North America and therefore our findings may have enhanced value in that region. We have made this explicit in the manuscript with the following lines:

L217-218: Agri-environment schemes and CRP aim to increase floral abundance and diversity in agricultural landscapes to benefit wider biodiversity

L222: effectiveness of schemes, such as AES and CRP, which aim to support biodiversity

L291-293. Finally, our results suggest that the CRP, which has a broadly similar approach to AES, could be tailored to combat emerging pollinator disease, as *N. bombi* is currently considered an emerging disease in North American bumblebee populations.

We have also added text to provide clarity on how caffeine forage could lead to nutritional deficiencies

L308-311: However, it should be noted that under field conditions caffeine consumption may lead to suboptimal foraging strategies, as caffeine-producing plants have been linked to an overestimation of forage quality in honeybees [71] which, if replicated in bumblebees, may have a negative impact on colony fitness through nutritional deficiencies.

Referee 2: *Line 52: font size changes*

Authors: Corrected as requested.

Referee 2: *Line 94: may benefit from some system specific references*

Authors: We agree and thank the reviewer for highlighting this and have included the following references:

[56] Lu, Y.H., Wu, C.P., Tang, C.K., Lin, Y.H., Maaroufi, H.O., Chuang, Y.C. et al. Identification of immune regulatory genes in *Apis mellifera* through caffeine treatment. *Insects*. 2020; 11: 516.

[57] Bernklau, E., Bjostad, L., Hogeboom, A., Carlisle, A., Arathi, H.S. Dietary phytochemicals, Honey bee longevity and pathogen tolerance. *Insects*. 2019; 10: 14.

Referee 2: *217-232: what about possible negative effects of phytochemicals?*

Authors: We agree that describing potential negative effects of phytochemicals is important and have included the following text

L295-296: Phytochemicals can negatively impact bumblebee fitness, for example, by reducing the production of sexual castes [65].

Referee 2: *254: clarify its not species specific within Bombus.*

Authors: Clarified as requested.

Referee 2: *271-272: are the planting schemes in the US?*

Authors: We have included the following text to add clarification

L 282-283: As such, planting strategies, such as AES or the North American equivalent CRP, that target incipient colonies may have a greater impact on pollinator parasite epidemiology

Referee 2: *307: what was the concentration of the inoculant? How was the inoculant also deemed clear of other parasites?*

Authors: We have included the final concentration of the inoculant as requested and have described how we ensured it was clear of other parasites:

L333-339: The resulting spore solution was centrifuged at 4°C for 10 minutes at 5000 rpm to isolate and purify the spore pellet. The pellet was resuspended in 100µl of 0.01M NH₄Cl and the inoculum was checked for the presence of non-target bumblebee parasites using a compound phase contrast microscope set to ×400 magnification. A larval *N. bombi* inoculant was prepared by combining inverted sugar water and pollen (3:1) to create an artificial worker feed as outlined in [66]. This was then combined in equal proportions (100µl:100µl) with the *N. bombi* inoculum to create an experimental inoculant which contained 50,000 spores/µl.

Referee 2: 308-309: *mention the use of UV irradiation to remove microbes, a necessary step for the experiment to ensure that the bees are only inoculated with the pathogen being assessed in the study. However, the sterilisation of different bee pathogens require different doses of irradiation - is it possible to include details on this or the supplier of the pollen.*

Authors: We have included the supplier of the pollen **L341** Biobest, Belgium. This supplier irradiates their pollen to remove a comprehensive suite of parasites (pers. comm.) – however, the exact details are confidential company data, and so we do not have access to them and cannot share them. In addition, we visually inspected a sample of the pollen under a contrast phase contrast microscope at x400 magnification to check for bumblebee parasites. This information has also been included in the methodology:

L341-344: All pollen, which came from mixed flowers (Biobest, Belgium), used in all experimental procedures was irradiated to remove any microbes and a sample of each pollen consignment was visually inspected to check for bumblebee parasites using a compound phase contrast microscope set to ×400 magnification.

Referee 2: 309: *was the pollen of a known flower? Was it tested for phytochemicals?*

Authors: Pollen was from mixed flowers, which was provided to experimental and control groups in all experiments. We have made this clear in the text and provided some context for why mixed pollen best suited our experimental paradigm, including an additional reference.

L 344-348: While pollen was not screened for phytochemicals it was provided to both control and experimental treatments across all experiments. Consequently, as pollen was from mixed flowers our methodology mimicked natural intake of pollen into bumblebee colonies [79] with the experimental addition of caffeine to sugar water representing the addition of sainfoin to this intake.

Reference: [79] Piko, J., Keller, A., Geppert, C., Batáry, P., Tschardtke, T., Westphal, C. et al. Effects of three follower field types on bumblebees and their pollen diets. *Basic Appl Ecol.* 2021. **52**: 95-108.

Referee 2: 315: *nice to see you checked repeatedly for adult parasite presence*

Authors: We would like to thank the reviewer for noting this, our experimental paradigm relies on infection-free bees so we wanted to be robust with our screening methodology.

Referee 2: 329: *to clarify, was caffeine placed on pollen at all?*

Authors: We opted not to place caffeine on pollen as within a solvent you can guarantee an equal distribution of phytochemical. Consequently, all bees had access to the same concentration of caffeine, which would allow for meaningful insights into the effect of caffeine on colony level infection. We have clarified this in the manuscript with the following

L 339-341 For all experimental procedures caffeine was administered to bumblebees via sugar water, caffeine was not administered to bees using pollen.

Referee 2: 330: *though you describe the volume of MeOH added to the sucrose, you haven't noted the volume of sugar water to interpret what the final concentration of the solvent was.*

Authors: We have noted that 4ml of MeOH was used as a solvent per litre of sugar water.

L370-371: Caffeine was added to sugar water using 4ml of 40% MeOH L⁻¹ as a solvent

Referee 2: 350: *For clarity, which workers were isolated to provide faecal samples – the recently eclosed workers or the original ‘nurse’ workers?*

Authors: We thank the reviewer for noting this and we have provided clarity in the manuscript with the following line:

L392: each newly eclosed worker’s faecal sample

Referee 2: 361-366: *good practice to isolate and recheck health like this*

Authors: We thank the reviewer again for complimenting our methodology and agree this is an important step, especially when using wild caught queens as any transmission may have occurred in a similar timeframe to catching and therefore potential infections need time to manifest before they can be accurately diagnosed.

Referee 2: 379-380: *those pesky queens!*

Authors: We are in total agreement with the reviewer 😊

Referee 2: 376: *it is not clear if the number of brood inoculated was controlled/standardised between colonies, and if not, was to total number of spores applied to the colony standardised? also was it just the L2/3 brood inoculated?*

Authors: The number of brood inoculated was controlled for in our statistical analysis and we can confirm that all available brood were inoculated at this point. We have confirmed this with the line:

L416: and the remaining incipient brood was inoculated with *N. bombi*

Referee 2: 383: *was there cat-litter or similar in the base to absorb faeces or could pools of faeces in the arena facilitate dispersal between foraging workers?*

Authors: We thank the reviewer for noting this point. We did not use absorbent material in the foraging arena as we found that foragers would return this to the nest, or be distracted with its presence, thus leading to non-typical foraging strategies. The cages we used had perforations in the base to allow for drainage. However, we would agree that dispersal could be facilitated in our foraging arenas and have included the following text in the discussion to reflect this:

L268-271: We would note that intracolony transmission may have been facilitated in our foraging arenas as these may have allowed faeces to accumulate. However, even with this potential interaction, prevalence was still significantly lower in our caffeine treated colonies.

Referee 2: 386: *Did this caffeine treatment include MeOH as before?*

Authors: We can confirm that it did, and we have made this clear in the methodology with the following text:

L423-424: experimental sugar water (as described above).

Referee 2: 431-432: *It would be nice to include these normality plots as supplementary*

Authors: We agree with the reviewer and have included them as plots in supplementary information (Folly et al 2021 PRSB Supplementary files).

Referee 2: *Some minor formatting inconsistencies;*

Authors: We thank the reviewer for identifying these.

Referee 2: *In line 520: '10' is placed in bold which is inconsistent with the rest of the formatting.*

Authors: Edited as requested.

Referee 2: *Lines 575-576: a grey background is included, which is inconsistent with other references.*

Authors: Edited as requested.